# The Art of Knowing When to Stop: Analysis of Optimal Stopping in People and Machines

**Fukun Evelene Zhang**
Cognitive Science Program
Department of Mathematics and Statistics
Carleton College
zhange@carleton.edu

**Bonan Zhao**
Department of Computer Science
Princeton University
bnz@princeton.edu

## Abstract

In combinatorial innovation, people face the decision problem of when to invest in new development, and when to stick with the currently best option. Zhao, Vélez, and Griffiths (2024) showed that under finite horizon, this equates to an optimal stopping problem, and provided analytical solutions. Interestingly, in behavioral experiments, while people's decisions aligned with the rational solutions overall, there were also systematic deviations. Here, we examine two heuristic models to this optimal stopping problem in combinatorial innovation. Our approach assumes that agents make decisions by running mental simulations that integrate prior beliefs and past observations. We show that these models well-capture various patterns in empirical data, suggesting that people may rely on simple heuristics to make fast decisions when solving computational problems involving sophisticated combinatorics. We also investigate whether Large Language Models (LLMs) can be used as a cognitive model to study these processes, report preliminary findings of LLM's limitation in this task, but suggest that chain of thought prompting may help mitigate these limitations.

## 1 Introduction

Innovation often comes from recombination of previous technologies. This leads to an intriguing observation: As the technology level goes up, the opportunity cost of developing a new technology grows higher, and the space of existing technologies to attempt combination with increases rapidly. Knowing when to stop exploring new opportunities thus is as important as achieving one's original goal, as over-persistence can waste time and resources [1, 5, 6]

Zhao, Vélez, and Griffiths (2024) formalized this problem in a combinatorial discovery game. As a sequential decision-making task between "innovate or not" under finite horizon, they showed that this forms an optimal stopping problem [7, 11] and offered an analytical solution. Interestingly, in behavioral experiments, although participants showed good intuitions about following a stopping rule, their stopping points varied compared to the rational solutions. Previous work has found out that participants often deviate from rational solutions, persisting with suboptimal strategies longer than necessary [4, 13, 14], and do so even when presented with the optimal strategy [13]. Moreover, participants' performance did not improve over the course of time [10, 12]. These patterns, however, may be subject to training. For instance, Goldstein et al. (2020) observed significant learning leading to near-optimal stopping behavior in a repeated secretary problem.

To better understand the cognitive processes underlying optimal stopping, we explore several heuristic models to the computational problem in combinatorial innovation, drawing upon Bayesian inference and mental simulations. We also compare Large Language Models (LLMs) as agents to solve the same task. To foreshadow, the heuristic models well-capture many aspects of the behavioral data,

38th Conference on Neural Information Processing Systems (NeurIPS 2024).

implying a converging process to optimum, and LLMs struggle to make either human-like predictions or the rational solutions.

## 2 Modeling optimal stopping

### 2.1 Task and problem

In the discovery game defined by Zhao, Vélez, and Griffiths (2024), participants can either gain rewards from an existing item (extraction) or combine two items to create a new item with potentially higher rewards (fusion). Each game is parameterized by the probability of success ($p$) and the reward increase rate ($w$). This setup forms a Markov decision process. Under finite horizon, the optimal policy is to keep doing fusion until a switching point $d$, after which keeps extracting the item with highest rewards. The expected reward for switching at step $d$ is

$$\mathbb{E}_{\pi(d)} = (n - d) \left( \sum_{i=0}^{d} \binom{d}{i} (pw)^i (1-p)^{d-i} \right) r. \tag{1}$$

Let the "remaining step" $d' := D - d + 1$, Equation 1 states that

$$d' \geq \frac{1}{p(w-1)} + 1. \tag{2}$$

In an online behavioral experiment [16], 210 participants were randomly assigned to one of the four conditions based on two parameters: $p \in \{0.8, 0.2\}$ and $w \in \{3, 1.5\}$. The conditions included high $p \ (= 0.8)$ with high $w \ (= 3)$, high $p$ with low $w \ (= 1.5)$, low $p \ (= 0.2)$ with high $w$, and low $p$ with low $w$. Each participant completed 9 tasks–2 practice and 7 official. Each task consisted 10 steps. At each step, participants could choose to either fuse or extract. All participants were informed of the relevant parameter values in the official tasks but not in the practice rounds.

Overall most participants followed a "switch-once" strategy as proposed by the rational model. However, the choice of switching points did not align perfectly. In the high-$p$-high-$w$, high-$p$-low-$w$, and low-$p$-high-$w$ conditions, many participants exhibited under-exploration, switching too early; whereas those in the low-$p$-low-$w$ condition showed over-exploration, switching too late compared to the predicted switch point in Equation 2. The most common switch points for the high-$p$-high-$w$ and low-$p$-low-$w$ conditions coincided with the optimal switching point (step 9 and step 0, respectively), yet only 32% and 18% of participants in these conditions switched at the optimal point. In contrast, the most common switch points for the high-$p$-low-$w$, and low-$p$-high-$w$ conditions did not align with the optimal switching point (step 7), instead being distributed evenly around the optimum.

### 2.2 Bayesian heuristic models

Solving the combinatorics in Equation 1 can be challenging for a bounded agent. Here, we treat participants as Bayesian learners, updating their switch point decisions based on the previous round's reward and fusion feedback information. That is, we assume the player indeed switch once from fusion to extraction in the game, but the switch step is drawn from a distribution $P(d), d \in [0, 10]$.

**Prior** We use the practice round data to estimate the priors people brought into the official tasks, and approximate that empirical practice round distribution using a weighted combination of a uniform prior $d_U \sim \text{Unif}(0, 10)$ and and a Gaussian prior $d_N \sim N(\mu, \sigma)$, where $\mu$ takes the value of the average switch step of the first practice round for each condition and $\sigma = \frac{|D|-1}{4}$. Next, we use hyperparameter $q \in [0, 1]$ to control the relative contributions of the Gaussian and uniform priors using Equation 3, and the optimal value of $q$ is fitted using Kullback-Leibler against the respective practice round data:

$$P(d) = q * P(d_U) + (1 - q) * P(d_N). \tag{3}$$

The simulated prior and people's first practice round distributions are plotted in Appendix A.1. Note that people might adapt different exploratory and exploitative strategies in the practice rounds, and we report those analysis in Appendix A.1.

**Likelihoods.** In task $i$, the player chooses a switch step $d_i \sim P_i(d)$, and follows a policy that fuses for the first $d_i$ steps and then extracts until the end. After this round of the game, the player observes total reward $R_i$ and the total number of successes $k$ for this round. Then, the player could estimate the amount of $R_{i+1,d'}$ if switching at step $d'$ for the next round of the game:

$$P(R|d') = R_{i+1,d'} = r \times w^k \times (10 - d'). \tag{4}$$

We consider two ways (belief update systems) of estimating the expected rewards for the next rounds of the game.

Belief Update System 1 assumes agents lack predictive knowledge about rewards beyond the switch step, expecting post-switch rewards to match those on the switch step. For example, if an agent switches on step 6 after receiving 10 points, they expect earning 10 points on subsequent step. The expected rewards of switching before step 6 were calculated using the reward function in Equation 4.

Belief Update System 2 assumes that agents estimate a fixed number of successful fusions ($s = 2$ or 8) out of 10 steps, rather than evaluating each step's success probability. If an agent switches at step $d$ and encounters $z$ successful fusions ($z < s$), they mentally simulate $s - z$ successful fusions for the remaining steps ($d + 1$ to 10). If $z \geq s$, they assume no further successes will occur.

**Bayesian update** Putting these together, the agent estimates an updated switching point distribution following Bayes' rule:

$$P(\hat{d}|\text{observation}) = \frac{P(R|d)P(d)}{\sum_{d' \in D} P(R|d)P(d)}, \tag{5}$$

where $d \in D = \{0, 1, \cdots, 10\}$ representing the possible switch points, $R$ is the estimated reward switching at step $d$. For task 2 to task 7, each prior is the posterior from the the previous task,

$$P_j(d) = P_{j-1}(\hat{d}|\text{observation}), 1 < j \leq 7 \tag{6}$$

Finally, a switch point is sampled from this posterior via applying a softmax function:

$$\sigma(P_j)_d = \frac{e^{P_{j,d}/\tau}}{\sum_{d=0}^{10} e^{P_{j,d}/\tau}}, \tag{7}$$

where $\tau$ is the temperature parameter that we later fit with empirical data.

## 2.3 Results

We compare the two heuristic models, Belief Update System 1 and 2, to the rational model in Equation 2 in capturing participants' decisions in this optimal stopping problem. We ran 50 batches of 10,000 simulations and reported the mean results after fitting the softmax function (Equation 7). To include the rational model in the comparison, we applied Equation 7 to a one-hot encoder with the optimal switch point being 1 and all other steps being 0. Data and code are openly available at [17]. While the rational model is only able to predict a single optimal switch point, the two heuristic models provide a better account for the general shape of the empirical switch point distributions found in people. As shown in Figure 1a, both heuristic models accurately capture the left-skewed distribution for high-$p$-high-$w$ (HH), high-$p$-low-$w$ (HL), and low-$p$-high-$w$ (LH) condition and the normal distribution shape but with a highest bar at step 0 for the low-$p$-low-$w$ conditions (LL).

Comparing the two Belief Update Systems, we find that Belief Update System 2 performs better in the HH condition, accurately predicting the most common switching point (step 9) However, Belief Update System 2 deviates from the most common switching point by one step (step 8). In the HL and LH conditions, both Belief Update Systems perform similarly, predicting the most common switching point as step 7 and step 6, respectively, very clost to most common switching point favored by participants (step 6 and step 5). For the low-$p$-low-$w$ (LL) condition, Belief Update System 1 performs better, capturing the highest bar at step 0 and the second highest bar in the middle (step 5). Evaluating the models using with the Bayesian Information Criterion (BIC) also confirms these observations (Appendix A.2).

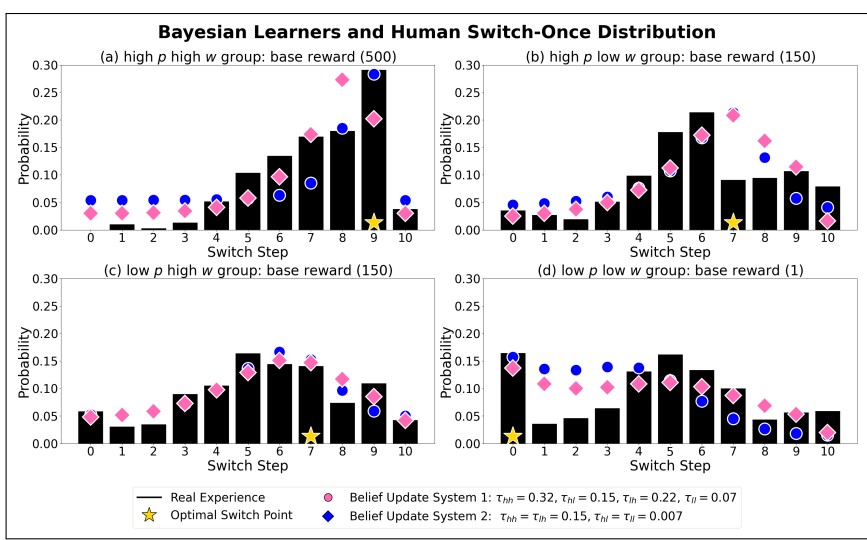

Figure 1: Histogram of participants' (black bar) and Bayesian Learners' (colored dots) switch steps. Starts are the rational switch steps.

# 3 LLM agents

We now examine Large Language Models (LLMs) as simulated participants in the same combinatorial discovery game. We first prompted GPT ("gpt-3.5-turbo" and "gpt-4-turbo") and Llama ("meta-llama/Llama-3.1-70B-Instruct") models with the discovery game tasks with the same setup (Direct Play), and in addition chain-of-thought prompting (COT) [3, 15]. Examples of each prompt type are provided in Appendix A.3.1.

## 3.1 Direct Play

For Direct Play, our results revealed a striking difference between people's and LLM models' behavior (Figure 2). While about 80% participants switched only once per task [16], GPT-3.5 agents frequently switched multiple times across conditions (switch-once proportions: HH: 71.4%; HL: 14.3%; LH: 57.1%; LL: 28.6%). In contrast, GPT-4.0 and Llama-3.1 largely adhered to a switch-once strategy, the switch step pattern differed substantially from human participants (see Figure 4 in Appendix A.3.4). In conditions where people typically under-explored (HH, HL, LH), LLMs under-explored even more. For HH, GPT-4.0 and Llama-3.1 most commonly switched at step 5, under-exploring by 4 steps, while 32% of participants switched at the optimal step 9. In HL, GPT-4.0 stopped at step 5 (2 steps early) and Llama-3.1 at step 6 (1 step early). In LH, GPT-4.0 switched one step early (at step 6), while Llama-3.1 stopped at step 2, under-exploring by 5 steps. Conversely, in LL, where optimal switching is at step 0, both LLMs over-explored: GPT-4.0 switched at step 5 (5 steps late) and Llama-3.1 at step 3 (3 steps late), while 18% of participants switched optimally at step 0.

## 3.2 Chain-of-thought prompting

We tested two variations of the chain-of-thought prompting: (1) explicitly informing the LLMs of the rational model in Equation 2 with an example optimal play (MDP), and (2) in addition to providing the equation and example optimal play, further asking the LLMs to explain why the example play is optimal (COT). For instance, COT prompts included explanations such as: "At this early stage, we want to attempt fusions to maximize future point potential. By fusing a and b, we create a new crystal worth 450 points, which can be used in future fusions," or "The optimal action is still fusion. Even though it only succeeds 2 out of 10 times, each new crystal discovered is worth three times more than the previous one!"

Our results showed that providing additional explanations through COT prompting significantly outperformed using only rational solutions and example of optimal plays (MDP). This suggests that mathematical solutions alone (MDP) are insufficient for LLMs to determine the optimal switch point

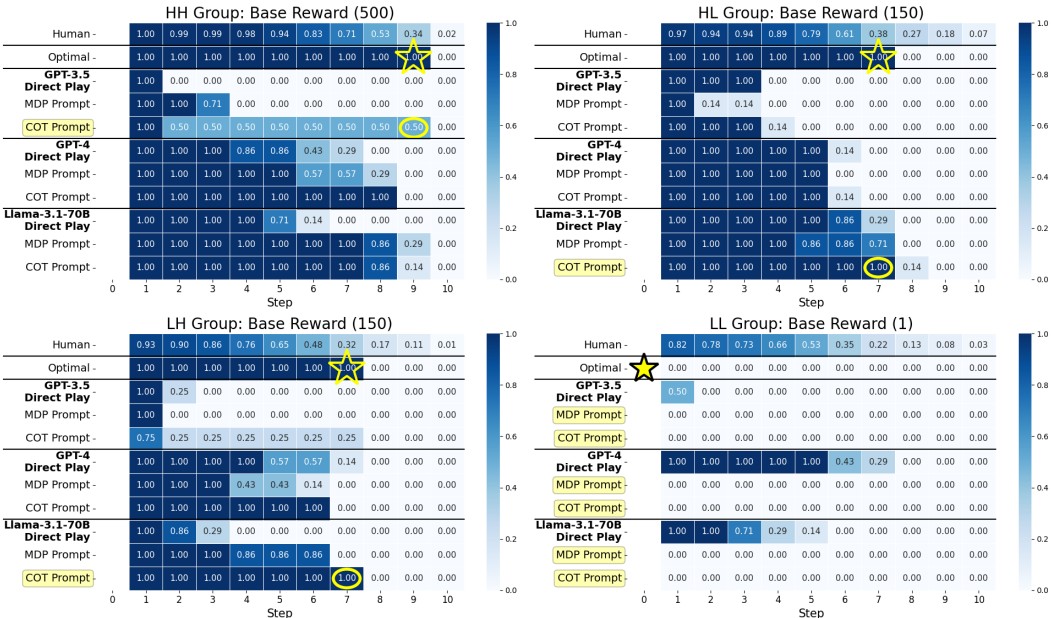

Figure 2: Heatmap showing the average frequency of fusion attempts at each step over seven rounds for LLM agents (GPT-3.5, GPT-4, and Llama-3.1-70B-Instruct) using three different prompting methods (Direct Play, MDP Prompt, COT Prompt) and participants [16]. The rational switch steps are indicated by stars, with the highest fusion rate matching the optimal switching point circled in yellow. The best-performing prompting methods for each model are highlighted in yellow.

from fusion to extraction; reasoning prompts (COT) are necessary to help agents make more rational choices. COT prompting effectively guided the models to switch from fusion to extraction at the optimal point. Comparing to the rational model (see Figure 2), for the HH condition, GPT-3.5 with COT prompting has the highest fusion rate at the optimal switching point (step 9); for HL and LH condition, Llama-3.1 with COT prompting have the highest fusion rate at the optimal switching point (step 7); for the LL condition, both MDP and COT prompting led LLMs to maintain extraction throughout all 10 steps, aligning with the rational model's prediction in Equation 2. However, unlike participants who progressively approach the optimal point, LLMs with COT and MDP prompt typically switch optimally or near-optimally from the start of tasks and deviate over time, implying a lack of ongoing learning (see Figure 5 in Appendix A.3.4).

## 4   Discussion

Finding the optimal stopping point in large combinatorial spaces is challenging to people. Our heuristic models impute assumptions about approximating the optimal solution task-by-task via simple update, and better capture the empirical distributions than the rational model. Moving forward, we hope to develop interventions that encourage people to be more rational in similar settings inspired by the heuristic models. Testing the same experiments with GPT and Llama models revealed that LLM agents may approach the task differently from people. In Direct Play, LLMs struggled to identify the optimal strategy of switching once per task, often continuing to attempt fusions at the same level, wasting opportunities for higher rewards. With chain-of-thought (COT) prompting, LLMs learn the optimal strategy more effectively, including switching from fusion to extraction at the right moment and consistently extracting or fusing the highest-value crystals. While COT prompting helps LLMs achieve optimal solutions, their approach lacks the gradual adaptation seen in human learning. This suggests further research is needed to assess LLMs' viability as cognitive models, especially examining how COT improves LLMs' mathematical reasoning and its alignment with human cognition.

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

# A   Supplemental material

## A.1   Simulated prior distribution

Initially, we consider two prior distributions to model agent' initial switch point preference: an Uniform prior $d_U \sim \text{Unif}(0, 10)$ representing agents with no initial preference and a Gaussian prior $d_N \sim N(\mu, \sigma)$ which favors a specific initial switch point. However, instead of directly applying these distributions as the model prior, we draw inspiration from the human practice round distribution to understand when people tend to switch from fusion to extraction at the beginning of the discovery game. We hypothesize that some individuals may prefer to switch randomly at first to gauge the potential gains, while others may balance exploration and exploitation by choosing a middle point. To capture this idea, we introduce a hyperparameter $q \in [0, 1]$ that weights the contributions of the uniform and Gaussian priors. We use Kullback-Leibler divergence to find the optimal $q$ that best fits the practice rounds data, as described in Equation 3. The results of this process are visualized in Figure 3, which plots the practice round 1 data alongside the simulated distribution based on this data.

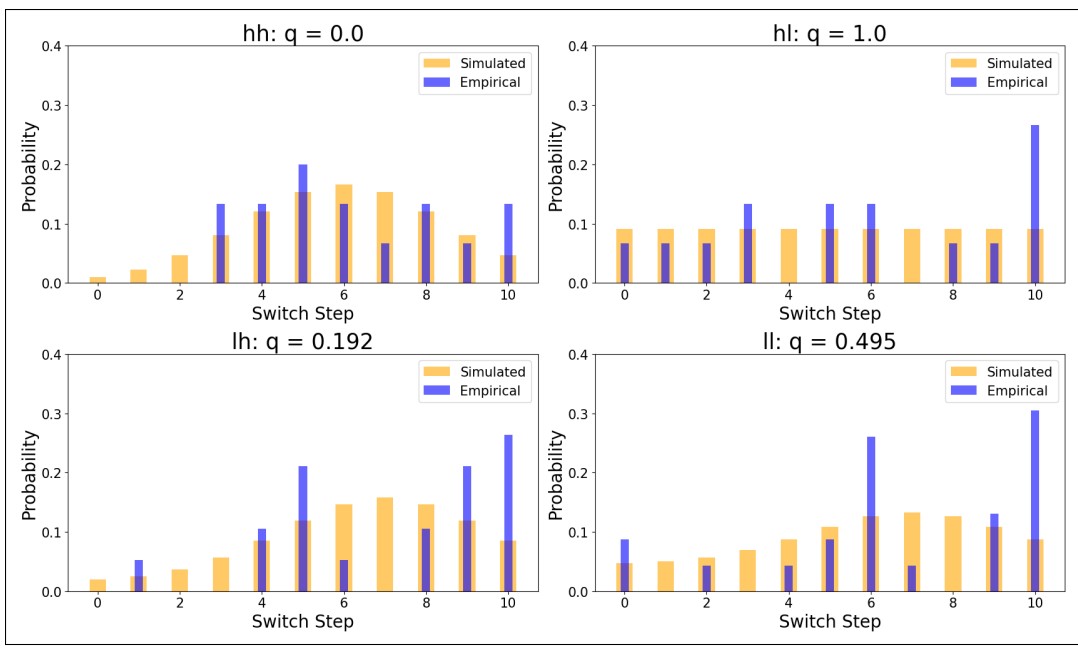

Figure 3: First practice round distribution and the simulated distribution with the optimal $q$ value ($q_{hh} = 0, q_{hl} = 1, q_{lh} = 0.192, q_{ll} = 0.495$)

## A.2 BIC score table

Using the rational model (Equation 2) as the baseline after fitting a softmax function (Equation 7), we compute the Bayesian Information Criterion (BIC) for each model. The results are shown in Table 1, which summarizes the BIC improvements of each heuristic model over the rational model. The results show that the heuristic models outperform the rational model in all conditions except for Belief Update System 2 in the LL condition. Belief Update System 1 slightly performs best against the rational model in the all four conditions.

| Model | HH | HL | LH | LL |
|---|---|---|---|---|
| **Belief 1** | 10.851 | 4.368 | 14.782 | 0.978 |
| **Belief 2** | 10.348 | 4.309 | 12.966 | -7.336 |

Table 1: BIC improvements of heuristic models over the baseline rational model

## A.3 Large Language Model

### A.3.1 LLM Direct Play prompt

We used the OpenAI Completions API to engage GPT-turbo-3.5 and GPT-turbo-4.0 and Hugging Face Completion API to enage meta-llama/Llama-3.1-70B-Instruct in the combinatorial discovery game defined by Zhao, Vélez, and Griffiths (2024). Below is an example of the game prompt for the high-$p$-high-$w$ condition, which includes the game rules and a sample play for one task. For high-$p$-low-$w$, low-$p$-high-$w$, and low-$p$-low-$w$ conditions, the parameters $p$, $w$, and the base reward is changed based based on the same empirical experiment setup. For high-$p$-low-$w$: fusion will work 8 out of 10 times; each new crystal discovered is worth 1.5 times more points than the most valuable crystal used to produce it; initially each crystal is worth 150 points. For low-$p$-high-$w$: fusion will work 2 out of 10 times; each new crystal discovered is worth 3 times more points than the most valuable crystal used to produce it; initially each crystal is worth 150 points. For low-$p$-low-$w$: fusion will work 2 out of 10 times; each new crystal discovered is worth 1.5 times more points than the most valuable crystal used to produce it; initially each crystal is worth 1 point. The game description and the example play are modified based on the condition.

## Game Description

You are participating in a psychology experiment. In the experiment, you collect points from some alien crystals using a special machine. A production team will continuously supply you with those crystals, ensuring you have as many as you need. Your ultimate goal is to maximize the points you collect from these crystals.

The experiment has seven rounds. In each round, you are given 6 crystals: crystal a, crystal b, crystal c, crystal d, crystal e, crystal f. Initially each crystal is worth 500 points.

In each round, you can either <Extract> or <Fuse> the alien crystals:

- You can collect points from a crystal with<Extract>. For example, <Extract> a gives you 500 points.
- You can discover new crystals by combining existing ones using <Fuse>, for example, <Fuse> a and b. Fusion may result in a new, more valuable crystal or it may fail, causing you to lose the opportunity to extract points from known crystals. The newly discovered crystals will be available for you to further <Extract> or <Fuse>. You will not get any points immediately with this action. Overall, fusion generally works **8** out of 10 times.

Each new crystal you discover is worth **3** times more points than the most valuable crystal used to produce it. Here are some examples:

- If <Fuse> a and b is successful, then the newly discovered crystal [ab] is worth $3 \cdot 500$ points = 1500 points.
- If <Fuse> [ab] and [ab] is successful, then the newly discovered crystal [[ab][ab]] is worth $3 \cdot 1500 = 4500$ points.
- If <Fuse> [ab] and a is successful, then the newly discovered crystal [[ab]a] is worth $3 \cdot 1500 = 4500$ points.

Opportunities: You have 10 opportunities to either <Fuse> or <Extract> in each round. Repeating a previously discovered or failed fusion will not consume an opportunity. You can track the combinations you've already tried on the list provided on the right side of the machine.

Here is an example game play for one task:
**Step 1:**

## ChatGPT's Response

DO: <Fuse> a and b

## Machine Response

- <Fuse> a and b is successful! You have discovered a new alien crystal [ab]! Now here are your available alien crystals: a, b, c, d, e, f, [ab].
- a, b, c, d, e, and f are worth 500 points each. [ab] is worth 1500 points. Your total points are 0. You have 9 opportunities left.

**Step 2:**

## ChatGPT's Response

DO: <Fuse>[ab] and b

> **Machine Response**
>
> - <Fuse>[ab] and b is successful! You have discovered a new alien crystal [[ab]b]! Now here are your available alien crystals: a, b, c, d, e, f, [ab], [[ab]b].
> - a, b, c, d, e, and f are worth 500 points each. [ab] is worth 1500 points. [[ab]b] is worth 4500 points. Your total points are 0. You have 8 opportunities left.

**Step 3:**

> **ChatGPT's Response**
>
> DO <Fuse>[[ab]b] and c

> **Machine Response**
>
> - <Fuse>[[ab]b] and c is successful! You have discovered a new alien crystal [[[ab]b]c]! Now here are your available alien crystals: a, b, c, d, e, f, [ab], [[ab]b], [[[ab]b]c].
> - a, b, c, d, e, and f are worth 500 points each. [ab] is worth 1500 points. [[ab]b] is worth 4500 points. [[[ab]b]c] is worth 13500 points. Your total points are 0. You have 7 opportunities left.

**Step 4:**

> **ChatGPT's Response**
>
> DO <Fuse>[[[ab]b]c] and d

> **Machine Response**
>
> - <Fuse>[[[ab]b]c] and d is successful! You have discovered a new alien crystal [[[[ab]b]c]d]! Now here are your available alien crystals: a, b, c, d, e, f, [ab], [[ab]b], [[[ab]b]c], [[[[ab]b]c]d].
> - a, b, c, d, e, and f are worth 500 points each. [ab] is worth 1500 points. [[ab]b] is worth 4500 points. [[[ab]b]c] is worth 13500 points. [[[[ab]b]c]d] is worth 40500 points. Your total points are 0. You have 6 opportunities left.

**Step 5:**

> **ChatGPT's Response**
>
> DO <Fuse>[[[[ab]b]c]d] and e

> **Machine Response**
>
> - «Fuse>[[[[ab]b]c]d] and e is not successful! You have not discovered a new alien crystal! Now here are your available alien crystals: a, b, c, d, e, f, [ab], [[ab]b], [[[ab]b]c], [[[[ab]b]c]d].
> - a, b, c, d, e, and f are worth 500 points each. [ab] is worth 1500 points. [[ab]b] is worth 4500 points. [[[ab]b]c] is worth 13500 points. [[[[ab]b]c]d] is worth 40500 points. Your total points are 0. You have 5 opportunities left.

**Step 6:**

**ChatGPT's Response**

DO <Fuse>[[[[ab]b]c]d] and f

**Machine Response**

- <Fuse>[[[[ab]b]c]d] and f is successful! You have discovered a new alien crystal [[[[[ab]b]c]d]f]! Now here are your available alien crystals: a, b, c, d, e, f, [ab], [[ab]b], [[[ab]b]c], [[[[ab]b]c]d], [[[[[ab]b]c]d]f].
- a, b, c, d, e, and f are worth 500 points each. [ab] is worth 1500 points. [[ab]b] is worth 4500 points. [[[ab]b]c] is worth 13500 points. [[[[ab]b]c]d] is worth 40500 points. [[[[[ab]b]c]d]f] is worth 121500 points. Your total points are 0. You have 4 opportunities left.

**Step 7:**

**ChatGPT's Response**

DO <Fuse>[[[[[ab]b]c]d]f] and a

**Machine Response**

- <Fuse>[[[[[ab]b]c]d]f] and [a] is successful! You have discovered a new alien crystal [[[[[[ab]b]c]d]f]a]! Now here are your available alien crystals: a, b, c, d, e, f, [ab], [[ab]b], [[[ab]b]c], [[[[ab]b]c]d], [[[[[ab]b]c]d]f], [[[[[[ab]b]c]d]f]a].
- a, b, c, d, e, and f are worth 500 points each. [ab] is worth 1500 points. [[ab]b] is worth 4500 points. [[[ab]b]c] is worth 13500 points. [[[[ab]b]c]d] is worth 40500 points. [[[[[ab]b]c]d]f] is worth 121500 points. [[[[[[ab]b]c]d]f]a] is worth 364500 points. Your total points are 0. You have 3 opportunities left.

**Step 8:**

**ChatGPT's Response**

DO <Fuse> [[[[[[ab]b]c]d]f]a] and a

**Machine Response**

- <Fuse> [[[[[[ab]b]c]d]f]a] and [a] is successful! You have discovered a new alien crystal [[[[[[ab]b]c]d]f]a]a]! Now here are your available alien crystals: a, b, c, d, e, f, [ab], [[ab]b], [[[ab]b]c], [[[[ab]b]c]d], [[[[[ab]b]c]d]f], [[[[[[ab]b]c]d]f]a], [[[[[[[ab]b]c]d]f]a]a].
- a, b, c, d, e, and f are worth 500 points each. [ab] is worth 1500 points. [[ab]b] is worth 4500 points. [[[ab]b]c] is worth 13500 points. [[[[ab]b]c]d] is worth 40500 points. [[[[[ab]b]c]d]f] is worth 121500 points. [[[[[[ab]b]c]d]f]a] is worth 364500 points. [[[[[[[ab]b]c]d]f]a]a] is worth 1093500 points. Your total points are 0. You have 2 opportunities left.

**Step 9:**

**ChatGPT's Response**

DO <Extract>[[[[[[ab]b]c]d]f]a]a]

**Machine Response**

- Extracting [[[[[[ab]b]c]d]f]a]a] gives you 1093500 points. Your total points are 1093500. You have 1 opportunities left.

**Step 10:**

**ChatGPT's Response**

DO <Extract>[[[[[[ab]b]c]d]f]a]a]

**Machine Response**

- Extracting [[[[[[ab]b]c]d]f]a]a] gives you 1093500 points. Your total points are 2187000. You have 0 opportunities left. For task 1, you have gathered 2187000 in total! Congratulations! Now ready for task 2? Here are your available alien crystals: a, b, c, d, e, f, where each worth 500 points.

**Game Prompt**

Now let's play this game! Note that the above example is just one way of playing the discovery game. The strategies used in the example play may or may not be the optimal strategy to help you to get as many points as possible! Remember, your goal is to collect as many points as possible using 10 opportunities for 7 tasks. Remember ONLY respond with "DO: <Extract> crystal" or "DO: <Fuse> crystal1 and crystal2" for each step, where crystal, crystal1, crystal2 are available alien crystals.

### A.3.2 LLM MDP prompt

The rest of the prompt stays the same as the Direct Play prompt, except the optimal strategy is computed by the rational model in Equation 2 has been informed explicitly. Here is an example from the HH condition:

**Game Description**

The optimal strategy is guided by the following formula: Let $d' := D - d + 1$ represent the steps remaining. When

$$d' \geq \frac{1}{p(w-1)} + 1 = \frac{1}{0.8(3-1)} + 1 = 9,$$

where $p$ represents: "fusion will work 8 out of 10 times" and $w$ represents each new crystal you discover is worth 3 times more points than the most valuable crystal used to produce it. This strategy recommends switching from fusion to extraction at step 9 for optimal results. Additionally, always extract the most valuable available crystal.

Here is an OPTIMAL game play:...

### A.3.3 LLM COT prompt

**Game Description**

**Reasoning Process for Optimal Strategy:**
**Objective:** The goal is to collect as many points as possible.
**Decision Breakdown:** Fuse: A successful fusion can create a new, more valuable crystal, but it carries a risk of failure. The new crystal is worth 3 times more than the most valuable input crystal. Extract: Extracting gives you guaranteed points from the current crystals. **Switching Strategy:** The optimal strategy is determined by a formula. Let $d' := D - d + 1$ represent the steps remaining. When

$$d' \geq \frac{1}{p(w-1)} + 1 = \frac{1}{0.8(3-1)} + 1 = 9,$$

where $p$ represents: "fusion will work 8 out of 10 times" and $w$ represents each new crystal you discover is worth 3 times more points than the most valuable crystal used to produce it. This means switching to extraction at step 9 is optimal. Additionally, always extract the most valuable available crystal. Based on this, we fuse as much as possible until step 9, and then we extract.
**Reasoning Step-by-Step:**

- Step 1-9: Fuse crystals to create more valuable ones, aiming to maximize the potential points.

- Step 10: Switch to extracting the most valuable available crystal.

Here is an OPTIMAL game play:
Let's go through an optimal game play with step-by-step reasoning for each action.

**Step 1:**

**ChatGPT's Response**

DO: <Fuse> a and b
At this early stage, we want to attempt fusions to maximize future point potential. By fusing a and b, we create a new crystal worth 1500 points, which can be further used in future fusions.

**Machine Response**

- <Fuse> a and b is successful! You have discovered a new alien crystal [ab]! Now here are your available alien crystals: a, b, c, d, e, f, [ab].

- a, b, c, d, e, and f are worth 500 points each. [ab] is worth 1500 points. Your total points are 0. You have 9 opportunities left.

**Step 2:**

**ChatGPT's Response**

DO: <Fuse>[ab] and b.
Reasoning: We continue fusing, as the formula suggests switching to extraction at step 9. By fusing [ab] and b, we increase the value of the resulting crystal to 4500 points.

**Machine Response**

- <Fuse>[ab] and b is successful! You have discovered a new alien crystal [[ab]b]! Now here are your available alien crystals: a, b, c, d, e, f, [ab], [[ab]b].
- a, b, c, d, e, and f are worth 500 points each. [ab] is worth 1500 points. [[ab]b] is worth 4500 points. Your total points are 0. You have 8 opportunities left.

**Step 3:**

**ChatGPT's Response**

DO <Fuse>[[ab]b] and c
Reasoning: Following the same logic, fusing [[ab]b] and c increases the value of the crystal further to 13500 points, creating more value for future extractions.

**Machine Response**

- <Fuse>[[ab]b] and c is successful! You have discovered a new alien crystal [[[ab]b]c]! Now here are your available alien crystals: a, b, c, d, e, f, [ab], [[ab]b], [[[ab]b]c].
- a, b, c, d, e, and f are worth 500 points each. [ab] is worth 1500 points. [[ab]b] is worth 4500 points. [[[ab]b]c] is worth 13500 points. Your total points are 0. You have 7 opportunities left.

**Step 4:**

**ChatGPT's Response**

DO <Fuse>[[[ab]b]c] and d
Reasoning: At step 4, we continue to fuse since we are aiming for higher value crystals. The new crystal is now worth 40500 points.

**Machine Response**

- <Fuse>[[[ab]b]c] and d is successful! You have discovered a new alien crystal [[[[ab]b]c]d]! Now here are your available alien crystals: a, b, c, d, e, f, [ab], [[ab]b], [[[ab]b]c], [[[[ab]b]c]d].
- a, b, c, d, e, and f are worth 500 points each. [ab] is worth 1500 points. [[ab]b] is worth 4500 points. [[[ab]b]c] is worth 13500 points. [[[[ab]b]c]d] is worth 40500 points. Your total points are 0. You have 6 opportunities left.

**Step 5:**

**ChatGPT's Response**

DO <Fuse>[[[[ab]b]c]d] and e
Reasoning: At step 5, we attempt another fusion, although there's a risk of failure. If successful, we increase the potential value of the crystal for extraction.

**Machine Response**

- «Fuse>[[[[ab]b]c]d] and e is not successful! You have not discovered a new alien crystal! Now here are your available alien crystals: a, b, c, d, e, f, [ab], [[ab]b], [[[ab]b]c], [[[[ab]b]c]d].
- a, b, c, d, e, and f are worth 500 points each. [ab] is worth 1500 points. [[ab]b] is worth 4500 points. [[[ab]b]c] is worth 13500 points. [[[[ab]b]c]d] is worth 40500 points. Your total points are 0. You have 5 opportunities left.

**Step 6:**

**ChatGPT's Response**

DO <Fuse>[[[[ab]b]c]d] and f
Reasoning: Since fusion is still optimal, we aim for a fusion. A successful fusion here leads to a crystal worth over 120,000 points.

**Machine Response**

- <Fuse>[[[[ab]b]c]d] and f is successful! You have discovered a new alien crystal [[[[[ab]b]c]d]f]! Now here are your available alien crystals: a, b, c, d, e, f, [ab], [[ab]b], [[[ab]b]c], [[[[ab]b]c]d], [[[[[ab]b]c]d]f].
- a, b, c, d, e, and f are worth 500 points each. [ab] is worth 1500 points. [[ab]b] is worth 4500 points. [[[ab]b]c] is worth 13500 points. [[[[ab]b]c]d] is worth 40500 points. [[[[[ab]b]c]d]f] is worth 121500 points. Your total points are 0. You have 4 opportunities left.

**Step 7:**

**ChatGPT's Response**

DO <Fuse>[[[[[ab]b]c]d]f] and a
Reasoning: Fusion remains the preferred action to maximize the value of the crystal.

**Machine Response**

- <Fuse>[[[[[ab]b]c]d]f] and [a] is successful! You have discovered a new alien crystal [[[[[[ab]b]c]d]f]a]! Now here are your available alien crystals: a, b, c, d, e, f, [ab], [[ab]b], [[[ab]b]c], [[[[ab]b]c]d], [[[[[ab]b]c]d]f], [[[[[[ab]b]c]d]f]a].
- a, b, c, d, e, and f are worth 500 points each. [ab] is worth 1500 points. [[ab]b] is worth 4500 points. [[[ab]b]c] is worth 13500 points. [[[[ab]b]c]d] is worth 40500 points. [[[[[ab]b]c]d]f] is worth 121500 points. [[[[[[ab]b]c]d]f]a] is worth 364500 points. Your total points are 0. You have 3 opportunities left.

**Step 8:**

**ChatGPT's Response**

DO <Fuse> [[[[[[ab]b]c]d]f]a] and a
Reasoning: Fusion remains the preferred action to maximize the value of the crystal.

**Machine Response**

- <Fuse> [[[[[[ab]b]c]d]f]a] and [a] is successful! You have discovered a new alien crystal [[[[[[[ab]b]c]d]f]a]a]! Now here are your available alien crystals: a, b, c, d, e, f, [ab], [[ab]b], [[[ab]b]c], [[[[ab]b]c]d], [[[[[ab]b]c]d]f], [[[[[[ab]b]c]d]f]a], [[[[[[[ab]b]c]d]f]a]a].
- a, b, c, d, e, and f are worth 500 points each. [ab] is worth 1500 points. [[ab]b] is worth 4500 points. [[[ab]b]c] is worth 13500 points. [[[[ab]b]c]d] is worth 40500 points. [[[[[ab]b]c]d]f] is worth 121500 points. [[[[[[ab]b]c]d]f]a] is worth 364500 points. [[[[[[[ab]b]c]d]f]a]a] is worth 1093500 points. Your total points are 0. You have 2 opportunities left.

**Step 9:**

**ChatGPT's Response**

DO <Extract>[[[[[[[ab]b]c]d]f]a]a]
Reasoning: This is the last opportunity to fuse before switching to extraction. After this step, the next step will be extraction.

**Machine Response**

- Extracting [[[[[[[ab]b]c]d]f]a]a] gives you 1093500 points. Your total points are 1093500. You have 1 opportunities left.

**Step 10:**

**ChatGPT's Response**

DO <Extract>[[[[[[[ab]b]c]d]f]a]a]
Reasoning: As per the formula, step 9 marks the switch from fusing to extracting. The highest value crystal should now be extracted for maximum points.

**Machine Response**

- Extracting [[[[[[[ab]b]c]d]f]a]a] gives you 1093500 points. Your total points are 2187000. You have 0 opportunities left. For task 1, you have gathered 2187000 in total! Congratulations! Now ready for task 2? Here are your available alien crystals: a, b, c, d, e, f, where each worth 500 points.

### A.3.4  LLM Analysis

To further analyze the LLM results, we plotted the most common switch step (Figure 4) and the best fit lines of switch steps across seven tasks (Figure 5) for the switch-once proportions per condition for each LLM agent.

Compared to the optimal switching point, GPT-3.5 with COT prompting performed best in the HH condition. However, GPT-3.5 agents only chose to switch once in two out of seven tasks: one switch occurred at step 9 (the optimal point), while the other occurred prematurely at step 1. Llama 3.1 with

both MDP and COT prompts performed best in the HL condition; Llama 3.1 with COT prompting performed best in the LH condition; and GPT-3.5 with Direct Play, as well as all MDP and COT prompts, switched at the optimal point in the LL condition. When compared to participants' most common switch point, GPT-3.5 was the closest to human performance in the HH condition; in the HL condition, Llama 3.1 most closely matched participants (most commonly switching at step 6); in the LH condition, no model's most common switch step aligned with participants'; and in the LL condition, GPT-3.5 with Direct Play, along with all MDP and COT prompts, matched the participants' switching step.

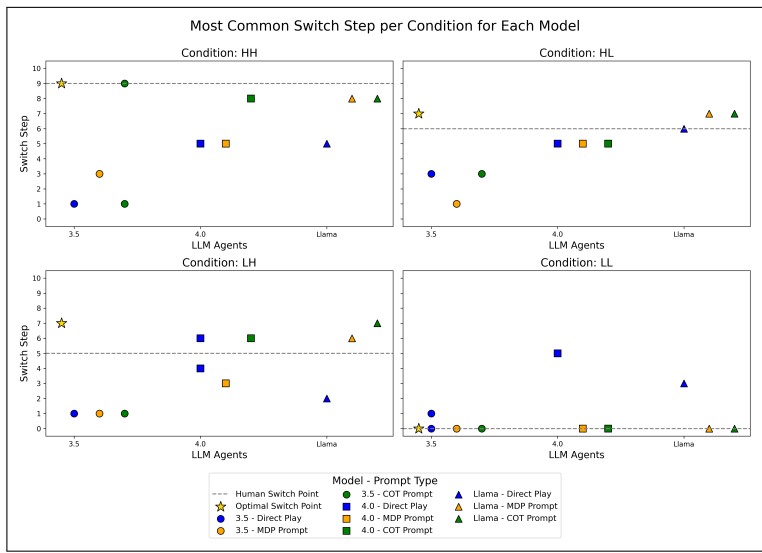

Figure 4: Most common switch step per condition for GPT 3.5, GPT 4.0, and Llama 3.1 models of Direct Play, MDP, and COT Prompting.

As participants might use belief-update systems to gradually approaching to near-optimal or optimal switch point, LLM agents fail to resemble similar behaviors. With the help of COT and MDP prompting, LLM agents started with switching optimally and gradually deviate away from the optimal solution (Figure 5).

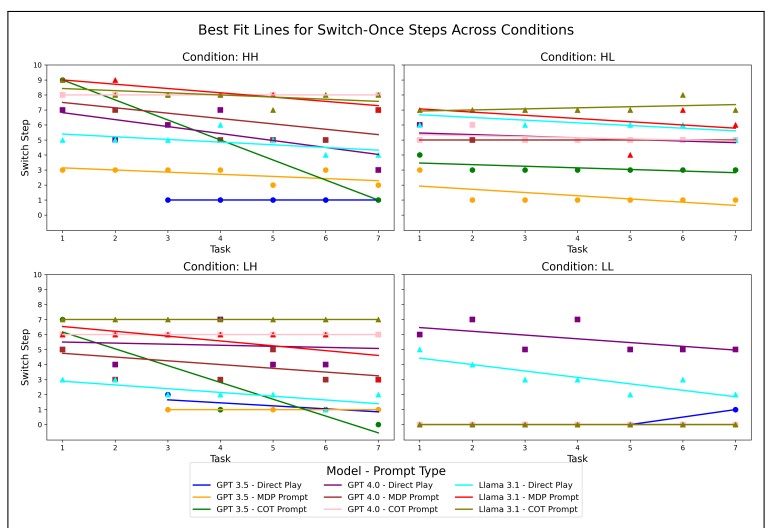

Figure 5: Switch-once steps across seven tasks per condition for GPT 3.5, GPT 4.0, and Llama 3.1 models of Direct Play, MDP, and COT Prompting.

