# OpenReview forum: "The Art of Knowing When to Stop: Analysis of Optimal Stopping in People and Machines"
_NeurIPS.cc/2024/Workshop/MATH-AI — MATH-AI 24_

### Official Review · Reviewer_kGQR · 2024-10-05
**This paper investigates optimal stopping in the context of combinatorial innovation, comparing human behavior with rational solutions and exploring heuristic models based on Bayesian inference and mental simulations. Additionally, it examines the performance of Large Language Models (LLMs) as agents in the same task. The authors find that while heuristic models effectively capture human decision-making patterns, LLMs struggle to replicate either human-like or rational behavior in this specific problem.**

**Rating:** 5
**Confidence:** 3

**Review:**

**Strengths:**

* **Novel and Interesting Research Question:** The paper addresses a relevant and intriguing question about human decision-making in combinatorial innovation, a domain with increasing practical importance.
* **Well-Defined Task and Formalization:** The combinatorial discovery game provides a clear and well-defined task for studying optimal stopping, allowing for a rigorous comparison between human behavior, rational solutions, and heuristic models.
* **Strong Empirical Evidence:** The paper presents compelling empirical data from human experiments, highlighting systematic deviations from rationality and providing a benchmark for evaluating heuristic models.
* **Effective Heuristic Models:** The proposed Bayesian heuristic models demonstrate a good fit to the empirical data, suggesting that people may rely on relatively simple cognitive processes to solve complex combinatorial problems.
* **Initial Exploration of LLMs as Cognitive Models:** The paper takes a preliminary look at LLMs as cognitive models for studying optimal stopping, highlighting some limitations of current LLMs in this domain.

**Weaknesses:**

* **Limited Scope of LLM Experiments:** The LLM experiments are relatively limited, focusing on a specific prompting technique and lacking a systematic exploration of alternative prompting strategies or different LLM architectures.
* **Lack of In-depth Analysis of LLM Behavior:** The paper identifies the limitations of LLMs in the optimal stopping task, but it does not provide a detailed analysis of the underlying reasons for these limitations or suggest specific strategies for improvement.
* **Limited Discussion of Broader Implications:** The paper could benefit from a more extensive discussion of the broader implications of the findings, particularly regarding the potential for developing interventions to improve human decision-making in combinatorial innovation and the future directions for research on LLMs as cognitive models.
* **Incorrect Reference Order:** The references are not listed in numerical order, which is a distracting formatting error that should be corrected.
* **Novelty of Heuristic Models Not Fully Articulated:** While the heuristic models show a good fit, the authors should elaborate on their novelty compared to existing decision-making models in cognitive science.
* **Preliminary LLM Evaluation:** The analysis of LLMs is limited to ChatGPT and lacks a systematic evaluation of other LLMs and the impact of prompt engineering.
* **Limited Theoretical Discussion:** The paper could be strengthened by exploring the theoretical implications of the findings, particularly regarding the nature of human heuristics in optimal stopping and their potential learnability.

**Recommendations:**

* **Expand LLM Experiments:** Explore a wider range of prompting techniques, including chain-of-thought prompting, and consider using different LLM architectures to gain a more comprehensive understanding of their capabilities and limitations in optimal stopping.
* **Analyze LLM Behavior in Depth:**  Investigate the reasons behind the observed LLM behavior, potentially through analyzing attention weights or intermediate representations, and propose strategies for improving LLM performance in this task.
* **Elaborate on Broader Implications:** Discuss the potential applications of the findings for developing interventions to enhance human decision-making in combinatorial innovation and other related domains.
* **Correct Reference Order and Formatting:** Ensure that the references are listed in the correct numerical order and adhere to the NeurIPS formatting guidelines.
* **Discuss Novelty of Heuristic Models:** Clearly articulate how the proposed Belief Update Systems relate to and differ from existing decision-making models in the cognitive science literature, highlighting their unique contributions.
* **Expand LLM Evaluation:** Conduct a more systematic evaluation of different LLMs, including those beyond ChatGPT, and investigate the impact of prompt engineering techniques on their performance in the optimal stopping task.
* **Explore Theoretical Implications:** Discuss the theoretical implications of the findings, such as whether human decision-making in optimal stopping can be effectively approximated by simple heuristics and how such heuristics might be learned or adapted.

**Overall Assessment:**

This paper presents a valuable contribution to the understanding of optimal stopping in combinatorial innovation. The rigorous experimental design, compelling empirical evidence, and effective heuristic models provide valuable insights into human decision-making processes. The initial exploration of LLMs as cognitive models also opens up interesting avenues for future research. **Although in its current state, the paper cannot be accepted due to the limitations outlined above, it is not that far off. I think either strengthening the llm analysis or removing the llm analysis from the scope of this paper and instead focusing on the bayesian methods might also be a viable alternative.** By addressing these limitations, including those highlighted in the major comments, and incorporating the suggested revisions, the paper can be significantly strengthened and potentially considered for inclusion in the NeurIPS MATH AI workshop.

**Minor Comments:**

* Page 1, Line 19: "increases exponentially" may be an overstatement. Consider rephrasing to "increases rapidly."
* Page 2, Line 52: Clarify what is meant by "official tasks" and "practice rounds."
* Page 3, Line 102: Consider providing a brief explanation of Bayesian Information Criterion (BIC) for readers unfamiliar with the concept.
* Page 4, Figure 1:  Label the x-axis in Figure 1b as "Step" for consistency with Figure 1a.

---

### Official Review · Reviewer_xhHv · 2024-10-07

**Rating:** 4
**Confidence:** 3

**Review:**

**Overall**:

This paper investigates the optimal stopping problem in a discovery game from two perspectives. First, the authors propose a Bayesian learning approach to model human decision-making based on behavioral experiment data. Then, they explore the performance of LLMs in solving the optimal stopping problem, offering potential directions for further research.

**Strengths**:

1. The paper is well-written and structured, with a clear problem definition and informative plots.

2. The simulation study offers detailed insights into which heuristic models may better explain human behavior regarding optimal stopping in the discovery game. Additionally, the comparison of the fusion rates between the LLM and human participants adds an intriguing dimension to the findings.

**Weaknesses**:

1. The goal of the paper seems somewhat unclear. The authors aim to model human behavior regarding switch points in a discovery game using two Bayesian heuristic models, comparing their performance in fitting data across different parameter values of $(p,w)$. Then, they assess LLM performance in the same context. While performance discussions are supported by plots, there is a lack of constructive reasoning or innovation based on these findings.

2. The paper's novelty may be limited. Bayesian models are widely used, and modeling human rationale with a relatively small sample size (210 samples) and comparing two specific modeling assumptions may not be sufficient for a significant contribution.


**Questions**:

1. In Lines 132-134, it’s suggested that using Bayesian heuristic models is somewhat standard in modeling human rationale for discovery games. While the results show promise for the LH and HH groups, the fit is less close for the HL and LL groups. How common is the use of Bayesian learners in discovery games, and could the authors cite more literature to support this?

2. In Figure 1 (right), the performance of the LLM agent appears highly unstable compared to human participants. How many replications were used to generate the fusion rate results for the LLM agent, and how do the authors explain the underlying reasons for this instability?

---

### Official Review · Reviewer_CRQt · 2024-10-08
**interesting area of using LLMs**

**Rating:** 7
**Confidence:** 3

**Review:**

LLMs are a type of artificial intelligence that are trained on massive amounts of text data, and they have shown impressive abilities to generate human-like text, translate languages, and answer questions in a comprehensive and informative way . The authors found that, LLMs do not provide a good model of human decision-making in this task . Specifically, the LLMs they tested failed to identify the optimal strategy of switching from exploration  to exploitation only once within a task, instead opting to find new combinations at the same level, even if it meant wasting opportunities to obtain higher rewards. The authors suggest that alternative prompting strategies might be needed to elicit more human-like behavior from LLMs in this task .
Overall, this paper makes several valuable contributions:
It provides insights into the cognitive processes that people use to solve optimal stopping problems.
It highlights the potential of using heuristic models to capture human decision-making in complex tasks.
It raises important questions about the limitations of current LLMs as cognitive models.
The findings of this paper have implications for a variety of fields, including cognitive science, artificial intelligence.

---

### Decision · Program_Chairs · 2024-10-09

Accept